# Designing Mid-Infrared Gold-Based Plasmonic Slot Waveguides for CO_2_-Sensing Applications

**DOI:** 10.3390/s21082669

**Published:** 2021-04-10

**Authors:** Parviz Saeidi, Bernhard Jakoby, Gerald Pühringer, Andreas Tortschanoff, Gerald Stocker, Florian Dubois, Jasmin Spettel, Thomas Grille, Reyhaneh Jannesari

**Affiliations:** 1Institute for Microelectronics and Microsensors, Johannes Kepler University, 4040 Linz, Austria; Bernhard.jakoby@jku.at (B.J.); Gerald.puehringer@jku.at (G.P.); reyhaneh.jannesari@jku.at (R.J.); 2Silicon Austria Labs GmbH, Europastr. 12, 9524 Villach, Austria; andreas.tortschanoff@silicon-austria.com (A.T.); florian.dubois@silicon-austria.com (F.D.); jasmin.spettel@silicon-austria.com (J.S.); 3Infineon Technologies Austria AG, Siemensstr. 2, 9520 Villach, Austria; gerald.stocker@infineon.com (G.S.); thomas.grille@infineon.com (T.G.)

**Keywords:** mid-infrared, plasmonics, slot waveguide, sensing applications, figure of merit

## Abstract

Plasmonic slot waveguides have attracted much attention due to the possibility of high light confinement, although they suffer from relatively high propagation loss originating from the presence of a metal. Although the tightly confined light in a small gap leads to a high confinement factor, which is crucial for sensing applications, the use of plasmonic guiding at the same time results in a low propagation length. Therefore, the consideration of a trade-off between the confinement factor and the propagation length is essential to optimize the waveguide geometries. Using silicon nitride as a platform as one of the most common material systems, we have investigated free-standing and asymmetric gold-based plasmonic slot waveguides designed for sensing applications. A new figure of merit (*FOM*) is introduced to optimize the waveguide geometries for a wavelength of 4.26 µm corresponding to the absorption peak of CO_2_, aiming at the enhancement of the confinement factor and propagation length simultaneously. For the free-standing structure, the achieved *FOM* is 274.6 corresponding to approximately 42% and 868 µm for confinement factor and propagation length, respectively. The *FOM* for the asymmetric structure shows a value of 70.1 which corresponds to 36% and 264 µm for confinement factor and propagation length, respectively.

## 1. Introduction

In recent years, surface plasmon polaritons (SPPs) have received much attention as they have opened up opportunities in many photonics applications. SPPs, which are guided electromagnetic waves, can be created when surface localized light waves along metal-dielectric interfaces are coupled to free electron oscillations in the metal [1,2]. Once they are excited, they allow for efficient light–matter interactions [3]. The maximum intensity of the resulting guided mode occurs at a metal/dielectric interface where the field amplitudes decay exponentially in the direction perpendicular to the interface [4]. This field, which is an evanescent field, indicates the bound, non-radiative nature of SPPs, preventing power from leaking away from the surface [5]. As SPPs can provide high field enhancement and break the diffraction limit associated with dielectric waveguides [2], they are of interest for photonic applications and particularly wave-guiding systems. The popularity of plasmonic waveguides is due to several features including high field intensity at the metal–dielectric interface and the capability of confining modes on the nanometer scale [6]. In addition, another merit of plasmonic waveguides is evident if sensors with surface functionalization are used, which guarantees the efficient interaction between the analyte and the mode in a region of high field intensity [7,8]. Moreover, plasmonic waveguides can be easily integrated into the optical circuits, where they show small footprints compared to fiber sensors [9]. Normally, light can be confined in both high and low refractive index waveguides [10,11]. However, when light is guided in the higher refractive index region, the interaction of light with the sensing medium will be restricted as the energy fraction contained in the evanescent field (containing the sensing medium) is small resulting in low sensitivity of the waveguide sensor [12]. Confining the light into the lower index region (containing the sample to be sensed) can improve the sensitivity of the waveguide. One of the best examples of guiding light in a low refractive index region is a slot waveguide, which has been used in many different applications [13,14,15,16]. The slot waveguide consists of a small gap with a low refractive index which is surrounded by two higher refractive index rails. The working principle of a plasmonic slot waveguide is based on the coupling of two modes confined at the edges of the two metal rails which surround the slot region. The associated modes are confined in the slot region [17] and a large amount of power that is confined there interacts with the sensing medium leading to a highly sensitive sensor [12].

Various materials have been introduced to act as plasmonic materials like metals, metal alloys, and semiconductors [18]. Among them, metals are widely used due to their high conductivity, large plasma frequency, and chemical stability, although they suffer from intrinsic losses [19]. Among metals, gold and silver are widely used for plasmonic applications owing to their relatively low loss in the visible and near-infrared (NIR) ranges. However, in terms of the fabrication processes, silver degrades fairly rapidly [18]. Therefore, in our case gold (Au) has been preferred due to its high resistance to oxidization compared to silver (Ag).

On the other hand, the mid-infrared (mid-IR) spectral area has attracted much attention as it includes absorption lines of several gases like NO, NH_3_, CO_2_, CO, CH_4_ [19,20,21,22]. The concentration of these molecules is an important indicator to assess the specific environmental conditions as well as medical disorders. Several gas sensors based on optical waveguides were designed and proposed, see e.g., [23,24]. In addition, some recent mid-IR waveguides based on CO_2_ sensing have been proposed and demonstrated [25,26,27]. In comparison to conventional dielectric waveguides, plasmonic waveguides typically can be expected to feature smaller propagation lengths (i.e., intrinsic damping), which are related to the intrinsic losses within the used metals. However, at the same time, by virtue of high evanescent field ratio (or confinement factor as used in this paper), the overall performance is comparable, as can be observed using proper figures of merit as will be shown below. The capability of plasmonic waveguides to maintain the electromagnetic field concentrated along the metal–dielectric interface leads to high confinement with a high intensity of electromagnetic field in a low index material representing the medium to be sensed (e.g., a gas). Depending on the considered application, this significant feature of plasmonic waveguides can compensate for the intrinsic ohmic losses leading to an improvement of the efficiency of the device. Among the plasmonic waveguides, the plasmonic slot waveguide provides lateral confinement and enables guiding of the wave as desired, which can also be an asset in the fabrication of miniaturized optical circuits. Therefore, in comparison to dielectric waveguides, they allow compact integration and short coupling lengths [28]. Plasmonic slot waveguides have been considered in the past [28,29,30,31]. They are particularly attractive as they allow the realization of a high-energy fraction guided outside of the waveguide material (i.e., the used metals) which, in turn, would result in high sensitivity with respect to absorption of a gaseous analyte in this region. In the present paper, we aim at systematic optimization for a fundamental plasmonic slot waveguide structure in order to identify typically feasible key properties.

We propose, design, and optimize two different free-standing and asymmetric gold-based plasmonic slot waveguides on a Si_3_N_4_ platform as shown in Figure 1. They are designed for mid-infrared sensing applications such as CO_2_ sensing. The first structure is a free-standing structure in which the gold rails are located on the top of the Si_3_N_4_ layer, and the upper and lower claddings are air (see Figure 1a), while the second structure is an asymmetric structure due to different materials used as upper and lower claddings (see Figure 1b). The free-standing plasmonic slot waveguide often supports a guided fundamental mode for any combination of the waveguide parameters, while an asymmetric one supports leaky modes above the cut-off slot width and slot height (metal thickness) [32].

Our sensor platforms are designed for a wavelength of 4.26 µm corresponding to the absorption peak of CO_2_ showing the typical application for IR-based gas sensing. As for our sensing application, two important factors, i.e., the confinement factor (Γ) and the propagation length, play important roles, we defined a figure of merit (*FOM*) which is obtained by multiplying the aforementioned key quantities. Thus, our goal is to design structures with a *FOM* as high as possible indicating high sensitivity. In addition, the mid-IR spectral region includes the absorption spectra of different gases such as CO and CH_4_, with absorption wavelengths at 4.6 µm and 3.92 µm, respectively. We merely adapted CO_2_ as an example to prove the concept. Yet, as our definition for the *FOM* is independent of the wavelength, it also works for the other wavelengths. However, the structures have to be optimized based on the selected wavelength.

The key components of a sensors system are a mid-infrared source, the plasmonic waveguide (interaction path), where the analyte interacts with the radiation, and a mid-infrared detector. A fully integrated mid-IR source, followed by a taper, can be used to couple light into the waveguide. The light source needs to be tailored considering the absorption band of the analyte. We note that the light coming from the infrared source can alternatively also be coupled into and out of the plasmonic waveguide through a grating structure [2]. Moreover, the limit of detection (LOD) is defined as the minimum concentration which can be detected by the sensor. The LOD depends both on the sensitivity of the sensor and the signal to noise ratio of the read-out system. For the signal to noise ratio, the measured signal from the sample with a known low concentration of analyte is compared to those of blank samples and by establishing the minimum concentration at which the analyte can be reliably detected. Therefore, as the LOD is crucially determined by system properties which are not determined by the sensor element itself, which is considered here, the LOD is not calculated in this work.

The rest of the paper is organized as follows: materials and methods are described in the Section 2. Results and discussions are presented in Section 3, and finally, we provide conclusions in Section 4.

## 2. Materials and Methods

The proposed waveguide geometries have been modeled using the finite element method (FEM) performed via commercially available COMSOL Multiphysics software. Modal analysis has been used for the cross-section of our waveguides. In order to minimize the spurious reflection at the boundaries of the simulation domain, the domain sizes were chosen to be sufficiently large. In addition, a scattering boundary condition was applied in our model to absorb outgoing waves and to prevent back reflection. Domains related to silica (*SiO*_2_), silicon nitride (*Si*_3_*N*_4_), and air (except the slot region) were meshed using a triangular mesh, while the rails and slot region were meshed using mapped elements. The grid sizes in the computational domains were selected in order to gain accurate results within the computational resources. The refractive index of *SiO*_2_ and *Si*_3_*N*_4_ at 4.26 µm are $n_{SiO_{2}}=$ 1.38 and $n_{Si_{3}N_{4}}$ = 1.88, respectively. The real and imaginary parts of the complex dielectric constants of gold have been assumed to be −848.18+62.499i [33].

## 3. Results and Discussion

### 3.1. Theoretical Concepts

#### 3.1.1. Confinement Factor (*Γ*)

Each gas has unique absorption lines which represent part of a specific fingerprint for the specific gas. Once the sensor platform is exposed to a gaseous medium which thus forms the cladding, the evanescent electromagnetic field (characterized by the evanescent field ratio (EFR), which describes the fraction of the guided electromagnetic field energy present in the evanescent field) begins to interact with the gas through absorption occurring in this region. Consequently, the transmitted light is attenuated if it corresponds to an absorption peak of that particular gas. According to the Lambert–Beer law [24], the decay in power through the sensor structure depends on the concentration of the gas under consideration, and the EFR of the waveguide. A sensor interacts more intensively with the absorbing medium when it shows a high EFR, and accordingly, the sensitivity increases. Two parameters affect the EFR of a specific mode: the dimensions of the waveguide with respect to the wavelength and the optical properties of the materials used for the waveguide and surrounding mediums [34]. The EFR is defined as the fraction of transmitted intensity in the air medium to the total modal intensity as shown below:(1)$$EFR=\frac{\iint_{Gas}^{}\varepsilon_{gas}\left(\|E⇀{\left(x,y)\right\|}^{2}\right)dxdy}{\iint_{All}^{}\varepsilon \left(x,y\right)\left(\|E⇀{\left(x,y)\right\|}^{2}\right)dxdy},$$
where $\vec{E}$ represents the electric field and ε is the permittivity of each material. The integrals cover the respective parts of the cross-sections which are assumed in the xy-plane. The integrands are proportional to the electromagnetic field density. We note that the packaging of the final device is planned such that the lower region (below the membrane) does not contain the analyte (also for mechanical stability).

For integrated waveguides, another parameter that should be taken into account is the group velocity which denotes the waveguide dispersion [35]. The group velocity (*V_g_*), which describes the speed at which the energy flows through a given cross-section of the waveguide, is explained in more detail in [36]. Therefore, confinement factor (Γ), which includes both EFR and *V_g_*, can be a reasonable measure for light–matter interaction. Γ is defined as [24]:(2)$$\Gamma =\frac{n_{g}}{n_{c}}\;EFR$$
where *n_g_* is the group index (*n_g_ = c*/*V_g_*, c is the speed of light in free space), and *n_c_* is the refractive index of the cladding. Combining both, the effects of field delocalization and dispersion can cause Γ to exceed unity, resulting in the facilitation of stronger absorption than obtained with a free-space beam [35,37].

#### 3.1.2. Propagation Length

The propagation wave vector of an SPP bounded mode at the metal/dielectric interface is given by [38]:(3)$$K_{SPP}=K_{re}+iK_{im}$$
where *K_re_* and *K_im_* are the real and imaginary parts of the *K**_SPP_*. Once an SPP mode is excited, a crucial parameter for the measurement is the propagation length [39]. The propagation length is the distance after which the intensity of a propagating SPP mode is reduced by 1/e. Since the intensity is proportional to the squared field quantities, the propagation length can be calculated by [38,40]:(4)$$L_{SPP}=\frac{1}{2K_{im}}$$

Unlike dielectric waveguides [41,42], whose propagation loss can be negligible, the existence of metal makes plasmonic waveguides lossy waveguides [43], where also scattering losses and leakage into the substrate contribute to the overall losses [24]. In reality, a thin adhesive layer (e.g., titanium) should be deposited under the gold. However, as such a thin layer would require a much finer mesh which leads to a very time-consuming calculation, it was not considered in our computations as its impact on the results can be expected to be negligible. Moreover, the loss originating from the surface roughness is not included.

The sensing mechanism is related to the application of the Beer–Lambert law to the waveguide sensors [35]:(5)$$I=I_{0}e^{-\alpha c\Gamma L}$$where *I* and *I*_0_ represent the measured intensity and the initial intensity, respectively, at a particular CO_2_ concentration *c*. Γ is confinement factor, α is the absorption coefficient and *L* is the interaction path, which in our case can be determined by calculating *L_SPP_*. Therefore, as both *L_SPP_* and Γ are crucial parameters for sensing applications, defining an appropriate *FOM* to deal with the trade-off between them is near at hand. For our goal, structures with maximum *FOM* showing favorable *L_SPP_* and Γ at the same time, are desirable. Hence, we define a dimensionless *FOM* as:(6)$$FOM=\Gamma \;\times \;L_{SPP}$$

The aforementioned *FOM* thus is a measure for the sensitivity of the waveguide. Our structures are optimized based on this *FOM* for CO_2_ sensing applications.

### 3.2. Free-Standing Structure

The free-standing plasmonic slot waveguide is composed of gold rails located on the thin *Si*_3_*N*_4_ layer with a thickness of 140 nm shown in Figure 1a. The 140 nm for silicon nitride thickness has been adopted based on some of our previous works with a different waveguide structure [44]. In fact, this thickness is the smallest thickness that is feasible using our technology. Therefore, we optimize our structures based on this thickness.

To start the optimization of the waveguide geometries, first the slot width is fixed to 200 nm. The calculated real part of the effective mode index (*N_eff_* = *K*_SPP_/*K*_0_, where *K*_0_ is the wave vector of free space light) and the *L_SPP_* of the fundamental mode as a function of rail width for different slot heights (H) are shown in Figure 2a,b, respectively. Increasing both rail width and slot height lead to the reduction of the real part of *N_eff_*. Moreover, increasing rail width up to 1000 nm leads to a rapid increase of the *L_SPP_*; further increasing the rail width leaves the *L_SPP_* almost unchanged. Moreover, increasing the slot height enhances the *L_SPP_*. However, the values of *L_SPP_* are virtually the same for slot heights above 500 nm especially, when the rail width exceeds 1000 nm. Thus, neither rail width nor slot height can influence the *L_SPP_* remarkably when they are above 1000 and 500 nm, respectively, because the fraction of energy transmitted into the gold layers remains approximately constant beyond the aforementioned values.

Figure 3a shows the Γ as a function of rail width for different slot heights. As one can see, less than 60% of the intensity is confined into the upper cladding and slot region (i.e., sensing medium) when the slot height is 100 nm, which corresponds to the lowest *L_SPP_* (see Figure 2b). However, more intensity can be confined in the slot area as the slot height increase, which leads to a higher Γ. Furthermore, with the increase of the rail width and considering that we have a thin layer of silicon nitride membrane, more field can penetrate the substrate and the nitride layer resulting in a reduced Γ. However, upon increasing the rail width beyond a value yielding a minimum of Γ again leads to an increase in Γ since the metal layer now starts acting as a shield keeping significant portions of the guided mode energy in the upper cladding (i.e., the air or the analyte in the sensing case). As the slot mode is laterally confined in the slot area, even further increasing the rail width has no significant effects on Γ and it approaches a virtually constant value.

The fraction of transmitted energy in gold rails, as indicated in Figure 3b, reduces with the increase of rail width leading to an increase of the *L_SPP_*. Nevertheless, it remains constant for slot height above 500 nm and the rail widths more than 1000 nm, which results in constant *L_SPP_*.

The *FOM* of the fundamental mode versus rail width for different slot heights is plotted in Figure 3c. The curves associated with different slot heights become close to each other as the slot height increases. No significant changes occur for slot heights beyond about 1000 nm, so this is considered to be the optimal value for the structure. In addition, a rapid increase is observed in *FOM* with increasing rail widths up to 1500 nm, and it is stabilized for further increasing of rail width. Therefore, the 1500 nm for the rail width is considered to be an optimized value.

Accordingly, keeping slot height and rail width to 1000 nm and 1500 nm, respectively, we aim to optimize the slot width of the free-standing structure. As illustrated in Figure 4a, the real part of *N_eff_* reduces as the slot width increases, which is the result of the weaker field confinement in the slot region [1]. Therefore, significant field intensity penetrates the substrate and cladding (see Figure 5 insets) which leads to a decreasing trend of *N_eff_* of the *SPP* mode. The *L_SPP_* increases with increasing slot width, due to the reduced absorption loss in the gold rails.

Figure 5 represents the Γ in sensing medium and the fraction of transmitted energy in the substrate (silicon nitride and air below) which are plotted by black and red curves, respectively. To provide better insight, absolute values of the lateral ($\left|E_{x}\right|$) and vertical ($\left|E_{y}\right|$) electric field components have been attached to the figure associated with slot widths of 100, 3000, and 5000 nm. The confinement is achieved through the metal strips surrounding the gap region and index guiding, respectively [45]. When the slot width is narrow (100 nm), most of the field (approximately 97%) is located in the medium to be sensed. Vertically, the SPP mode is confined at the gold/air and gold/*Si*_3_*N*_4_ interfaces. For this slot width, only around 4% of the field intensity penetrates the substrate, which indicates that the leakage associated with this mode is fairly limited. As slot width becomes wider, the Γ starts to decrease, yet *L_SPP_* increases (see Figure 4b) which can be attributed to the lower field intensity in the metallic regions. When increasing the slot width, the mode increasingly disintegrates into two modes confined at the edges of the two metal rails. As shown in Figure 5, for the slot width of 5000 nm, the SPP slot mode coupling vanishes, and the waveguide starts to act like two separate strip waveguides. Therefore, it can be concluded that the wider gap region leads to lower Γ and yet higher *L_SPP_*.

The results discussed above mean that a slot with larger width has opposite effects on *L_SPP_* and Γ. Thus, obtaining a tradeoff between Γ and *L_SPP_* is necessary for sensing applications. As shown in Figure 6, the maximum value for *FOM* is 274.6 corresponding to an optimal slot width of 3000 nm. This *FOM* equals to the highest Γ (roughly 42%) and *L_SPP_* (868 µm) we can achieve simultaneously. In addition, the field enhancement in the slot region for the optimized geometries of the free-standing plasmonic waveguide is 142%.

The effect of *Si*_3_*N*_4_ thickness on the *FOM* of the optimized structure has been investigated. As indicated in Figure 7a; the thinner the *Si*_3_*N*_4_ layer, the higher the *FOM*. The corresponding Γ and *L_SPP_* as a function of *Si*_3_*N*_4_ thickness are depicted in Figure 7b. As the *Si*_3_*N*_4_ thickness becomes thinner, less field energy can be absorbed by the *Si*_3_*N*_4_ layer resulting in higher *L_SPP_*. In addition, more field energy will be confined in the air region leading to higher Γ. For example, for a *Si*_3_*N*_4_ thickness of 100 nm, an Γ of 46% and a *L_SPP_* of 1 mm are feasible. However, the structures combining 1000 nm metal layers and 100 nm *Si*_3_*N*_4_ suspension layer which lead to a maximum propagation length of 1 mm constitute a challenge for the fabrication. Stress engineering and reduction in the sense of structuring the nitride layer outside the device area [46] or substitution of *Si*_3_*N*_4_ by low-stress *SiNx* thin film can be considered [47].

### 3.3. Asymmetric Structure

Before the optimization of the asymmetric structure, we investigate the influence of changing the substrate’s refractive index (*n_sub_*) on the *FOM* of the optimized free-standing structure. We note that *n_sub_* refers to the refractive index of the region below the *Si*_3_*N*_4_ layer which is air and silica for the first and second structures, respectively. As can be seen in Figure 8a, the *FOM* decreases rapidly to 46.7 when *n_sub_* approaches the refractive index of silica (*n_sub_*
≈ 1.4). Increasing *n_su_*_b_ leads to further field penetration into the substrate resulting in an Γ of around 5% which is shown in Figure 8b. Therefore, the high refractive index mismatch between the upper (*n* = 1) and lower cladding (*n_sub_* = 1.4) will result in less Γ and yet higher *L_SPP_* (see Figure 8b). Figure 8c represents the lateral and vertical components of the mode when *n_sub_* changes from the air (*n_sub_* = 1) to silica (*n_sub_* = 1.4). As clearly shown in Figure 8c, for *n_sub_* = 1, the vertical confinement ($\left|E_{y}\right|$) is more or less symmetric. However, the symmetry vanishes for *n_sub_* = 1.4 due to the high refractive index mismatch between upper and lower claddings, and the mode will leak significantly into the silica substrate. Furthermore, increasing *n_sub_* leads to a reduced intensity of the lateral field component ($\left|E_{x}\right|$) and eventually vanishing coupling between the two edge modes in the slot region. Hence, the optimal parameters for the first structure do not apply to the second structure because in this case, only two separate strip waveguide modes exist. Therefore, the dimensions of the waveguide have to be more compact to form the slot mode and prevent leakage (or at least obtain less leakage) to the substrate when we deal with an asymmetric slot waveguide.

The asymmetric structure as represented in Figure 1b is the one in which the gold-based slot waveguide is located on the top of the *Si*_3_*N*_4_ layer while both layers supported by *SiO*_2_ substrate. In practice, this substrate can be an oxidized silicon wafer. As already mentioned, this asymmetrical structure supports leaky modes above the cutoff slot width and height [32], while the free-standing structure as its *N_eff_* show (see Figure 2a and Figure 4a) supported guided mode for any combination of waveguide geometries without any cut-off. More specifically, when slot width approaches gradually to the cutoff slot width, the modal intensity pattern changes such that more field energy is contained in the neighboring substrate region and hence the effective refractive index approaches that of silica. This will eventually lead to leakage into the substrate when increasing the slot width further beyond cutoff. Figure 9a shows the real part of *N_eff_* of the fundamental mode as a function of slot width for different slot heights, while the rail width is fixed at 200 nm. The horizontal dashed line shows the refractive index of silica ($n_{SiO_{2}}$) and the three black vertical dashed lines represent the cutoff slot widths (*W_c_*) for slot heights of 300, 360, and 400 nm which are 240, 140, and 120 nm, respectively. For *N_eff_* larger than n_*SiO*2_, the electric field is tightly confined in the slot region laterally. When *N_eff_* is approaching $n_{SiO_{2}}$, the related fields extend more into the upper cladding and substrate. Below $n_{SiO_{2}}$, the mode will be leaky. We indicate the leaky mode with the dashed line using asterisks. Therefore, we only consider effective mode indexes, which are above the dashed line and below the *W_c_* representing the guided mode. The *L_SPP_* of the fundamental mode versus slot width for different slot height is represented in Figure 9b. The mode confinement decreases with increasing slot widths, leading to less fraction of intensity in the gold rails. Therefore, the *L_SPP_* increases as shown in Figure 9b. Similarly, for a given slot width, the *L_SPP_* increases as slot height increase due to the same reason which is mentioned for the slot width. The *L_SPP_* will increase even when the mode becomes leaky as clearly observed in Figure 9b. This is due to the fact that the dominant loss mechanism is the material loss in the metal [32].

Moreover, as slot width increases, the mode becomes less confined in the sensing medium and penetrates the substrate. In addition, confinement in the gold rails reduces as slot width increases. As a result, Γ decreases as plotted in Figure 10a. Based on the calculated results shown in Figure 10b, the optimal *FOM* is 63.5 corresponding to the slot width and height of 400 and 260 nm, respectively.

Using the optimal values for slot width and height, the optimization of *FOM* versus rail width was performed and the calculated results are depicted in Figure 11a. The maximum *FOM* is 70.1 corresponding to the optimal value of 700 nm for rail width representing 36% and 264 µm for Γ and *L_SPP_*, respectively, as plotted in Figure 11b. Comparison of the *FOM* obtained in Figure 8a for $n_{sub}=1.4$ shows that the optimization of the asymmetric structure improves the *FOM* from 46.7 to 70.1. For this structure, the field enhancement in the slot region for the optimized parameters is 509%. Furthermore, the distribution for the absolute values of the x and y components of electric fields are shown in Figure 11c for the optimized parameters. The mode is laterally confined in the slot region and vertically confined at the gold/air and gold/*Si*_3_*N*_4_ interfaces, although part of the mode extends to the upper cladding and substrate. The comparison between the profile of the electric field intensity ($\left|E_{x}\right|$ and $\left|E_{y}\right|$) in Figure 8c and the electric field components of the asymmetric structure (see Figure 11c) shows that the optimization yields guided mode featuring characteristic slot mode patterns.

Although the Γ of the free-standing and asymmetric structures, which is investigated in this paper, are approximately the same as that calculated for free-standing silicon waveguide in [25], the FOMs obtained for our structures are lower than that reported in [25]. Yet, the fabrication of our structures is technologically less challenging and more robust. In comparison with the dielectric waveguides designed in [34,44], our structures show lower FOMs. Nevertheless, our free-standing and asymmetric plasmonic waveguides indicate EFRs as high as 37% and 24.5% respectively, which are higher than EFRs reported for waveguides in [34,44]. In addition, the propagation length of the investigated free-standing plasmonic waveguide is more than 800 µm, which is a relatively long propagation length for a plasmonic waveguide. Moreover, the significant field enhancement in the slot area of free-standing and asymmetric plasmonic waveguides which are 142% and 509%, respectively, makes them attractive as potential candidates for some applications such as non-linear optics and light manipulations in a nanoscale waveguide. Finally, the fabrication of the presented structures is simple and complementary metal-oxide-semiconductor (CMOS) compatible.

## 4. Conclusions

The aim of this paper is to provide a quick insight into the opportunities provided by plasmonic slot waveguides. Focusing on silicon nitride as a platform as one of the most common materials systems, we proposed, designed, and optimized two gold-based plasmonic slot waveguides for mid-infrared CO_2_ sensing applications. Our waveguide geometries were optimized based on the figure of merit which is defined as the product of confinement factor and propagation length. For the free-standing plasmonic waveguide which always supports a bound SPP mode, neither rail width nor slot height beyond 1000 and 500 nm, respectively, can impact the propagation length, because the energy transmitted in the gold layer is virtually constant. Furthermore, above the 1500 and 1000 nm for rail width and slot height respectively, the *FOM* saturates. The optimal *FOM* for optimized geometries of free-standing structure is 274.6 and corresponds to approximately 42% and 868 µm for confinement factor and propagation length, respectively. Decreasing the silicon nitride thickness can improve the *FOM* but at the expense of more challenging fabrication. The asymmetric structure with a defined rail width indicates a guided mode only below the cut-off slot width and cut-off slot height. Above the cut-offs, the mode will be leaky. For the rail width of 200 nm, the cut-off slot width and height are 400 nm and 260 nm, respectively. The propagation length increases with the increase of slot width. The best *FOM* for the optimized parameters of asymmetric structure is 70.1 and indicates 36% and 264 µm for confinement factor and propagation length, respectively. This *FOM* is lower than that obtained for the free-standing structure however, in terms of fabrication, removing the substrate can lead to issues, e.g., in terms of mechanical stability.

## Figures and Tables

**Figure 1 sensors-21-02669-f001:**
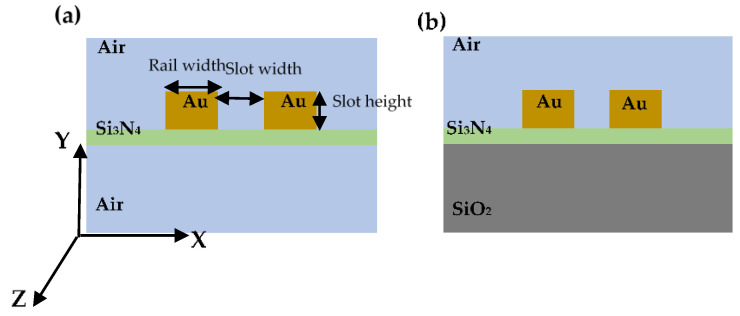
Schematic illustrations of the cross-section of two different gold-based plasmonic slot waveguides. (**a**) The gold rails are supported by a thin *Si*_3_*N*_4_ layer (free-standing structure). (**b**) The gold rails are located on the top of a thin *Si*_3_*N*_4_ layer where both layers are supported by a *SiO*_2_ substrate (asymmetric structure).

**Figure 2 sensors-21-02669-f002:**
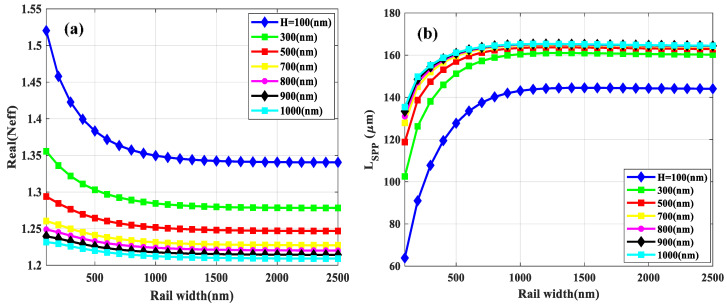
(**a**) The real part of the effective mode index (*N_eff_)* and (**b**) the *L_SPP_* versus rail width for different slot heights (H). The slot width is fixed at 200 nm.

**Figure 3 sensors-21-02669-f003:**
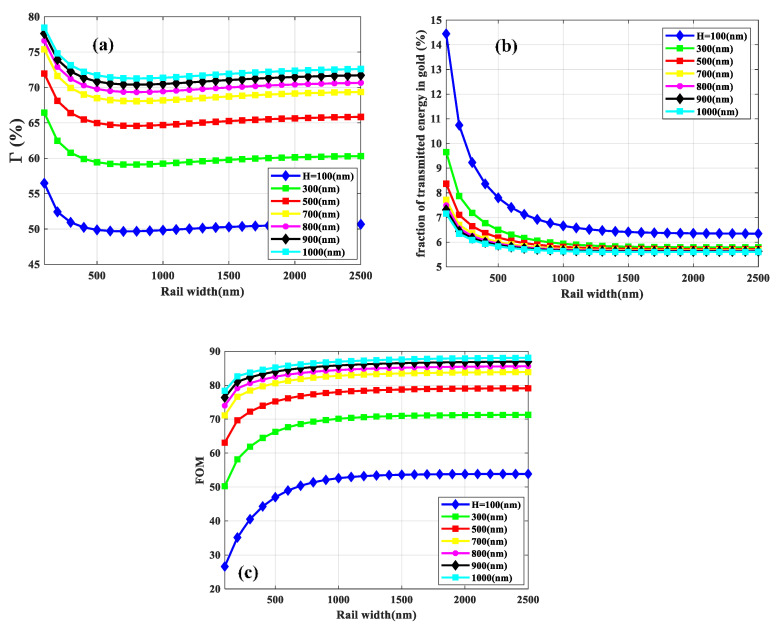
The confinement factor (Γ) in sensing medium (**a**), the fraction of transmitted energy in gold rails (**b**) and the *FOM* (**c**) of the fundamental mode versus rail width for different slot heights (H).

**Figure 4 sensors-21-02669-f004:**
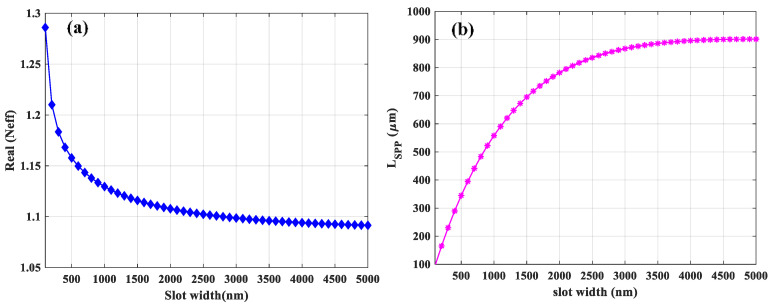
The real part of the effective mode index (*N_eff_)* (**a**) and *L_SPP_* (**b**) of the fundamental mode as a function of slot width. The slot height and rail width are 1000 and 1500 nm, respectively.

**Figure 5 sensors-21-02669-f005:**
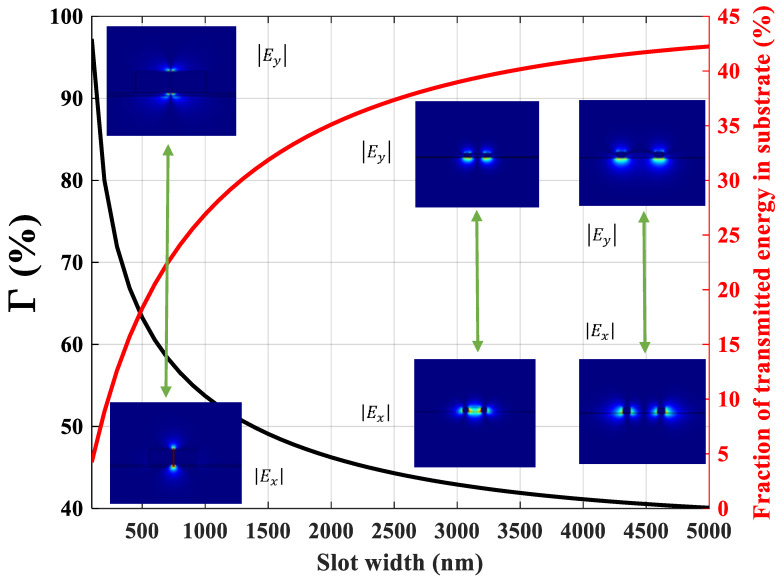
The confinement factor (Γ) indicating the amount of intensity confined in the sensing medium (black curve) and the fraction of transmitted energy in the substrate (red curve). The insets show the absolute values of $\left|E_{x}\right|$ and $\left|E_{y}\right|$ components of electric filed for slot widths of 100, 3000, and 5000 nm. The arrows relate two components of the electric field distribution of each point to each other.

**Figure 6 sensors-21-02669-f006:**
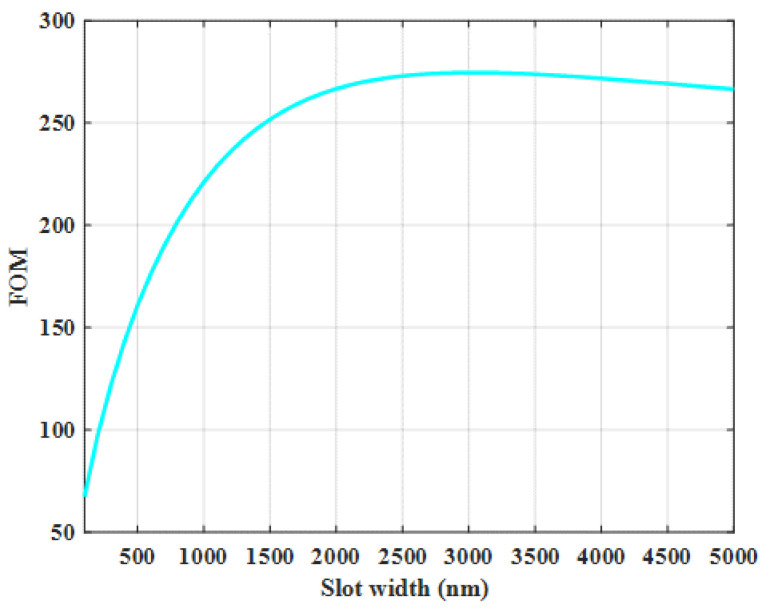
The *FOM* as a function of slot width. Slot height and rail width are 1000 and 1500 nm, respectively.

**Figure 7 sensors-21-02669-f007:**
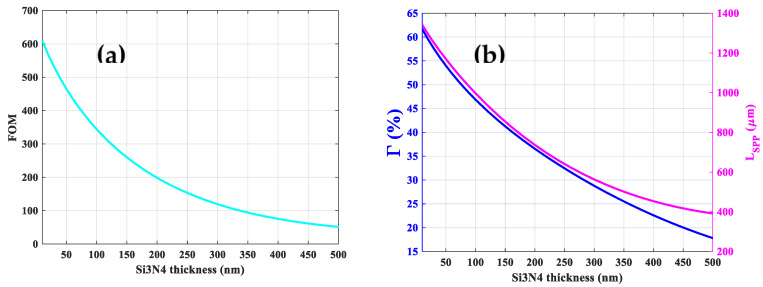
Figure of merit (*FOM*) (**a**) and corresponding Γ and *L_SPP_* (**b**) as a function of *Si*_3_*N*_4_ thickness for the optimized structure.

**Figure 8 sensors-21-02669-f008:**
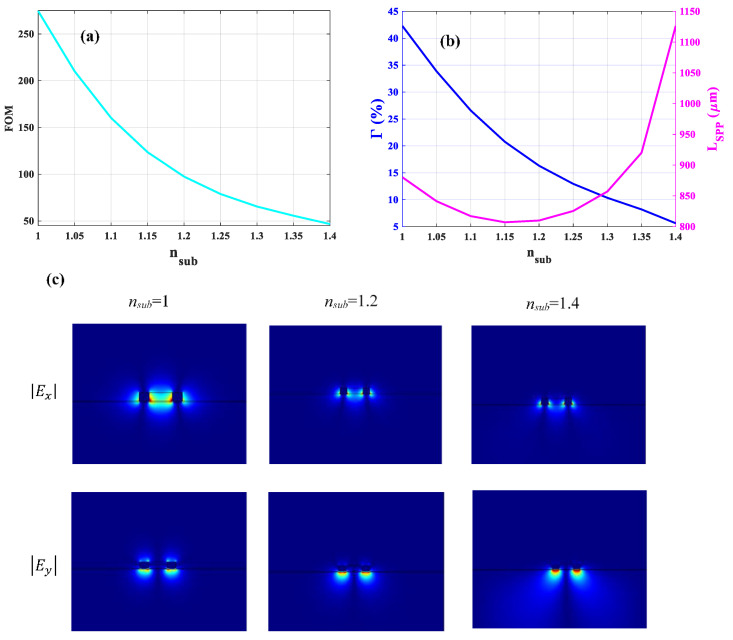
(**a**) Variation of *FOM* of the optimized symmetric structure with respect to the refractive index of the substrate (*n_sub_*). (**b**) The corresponding Γ (y-axis to the left) and *L_SPP_* (y-axis to the right). (**c**) Absolute values of x ($\left|E_{x}\right|$) and y ($\left|E_{y}\right|$) components of electric field distribution for different *n_sub_*.

**Figure 9 sensors-21-02669-f009:**
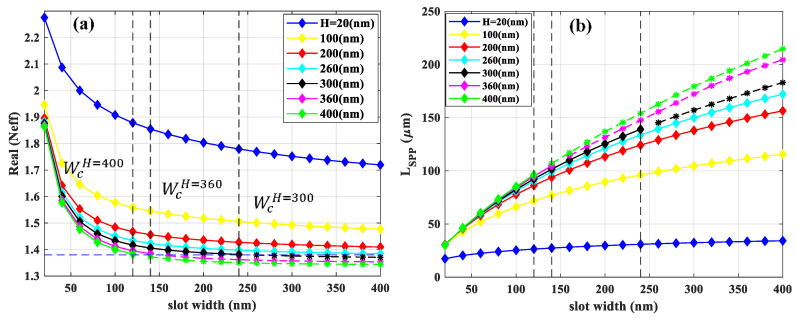
(**a**) The real part of effective mode index (*N_ef_*_f_) and (**b**) *L_SPP_* of the fundamental mode as a function of slot width for different slot heights. The horizontal dashed line indicates the refractive index of silica. The rail width is fixed at 200 nm. The three black vertical dashed lines show the *W_c_* for H = 300, 360, and 400 which are 240, 140, and 120 nm, respectively. The leaky modes are shown with dashed lines with asterisks.

**Figure 10 sensors-21-02669-f010:**
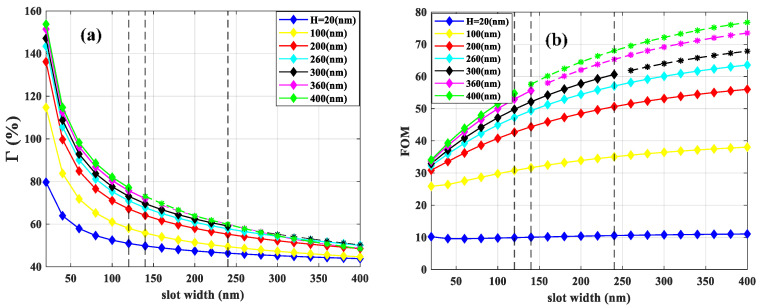
(**a**) The Γ and (**b**) *FOM* as a function of slot width for different slot heights. The three black dashed lines show the *W_c_* for H = 300, 360, and 400 which are 240, 140, and 120 nm, respectively.

**Figure 11 sensors-21-02669-f011:**
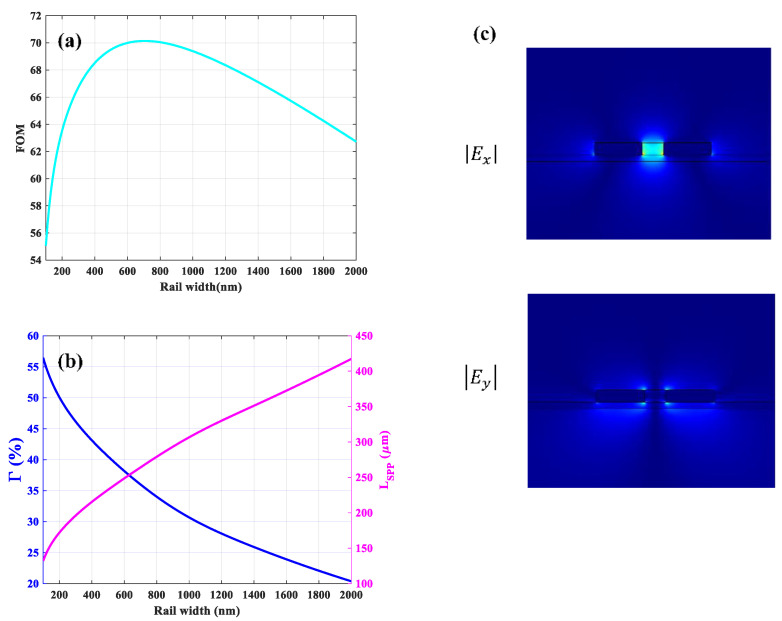
(**a**) The *FOM* and (**b**) corresponding Γ and *L_SPP_* as a function of rail width. The slot width and height are 400, and 260 nm, respectively. (**c**) The absolute values of x ($\left|E_{x}\right|$) and y ($\left|E_{y}\right|$) components of electric field distribution for the optimized structure.

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
