# Peer review of "Designing Mid-Infrared Gold-Based Plasmonic Slot Waveguides for CO2-Sensing Applications"

_sensors, 2021, doi:10.3390/s21082669_

Round 1
Reviewer 1 Report
The manuscript “Designing Mid-infrared Gold-based Plasmonic Slot Waveguides for CO2 Sensing” by Parviz Saeidi et. al. proposes, designs, and optimizes two different symmetric and asymmetric gold-based plasmonic slot waveguides on a Si3N4 platform for CO2 gas sensing. In general, the simulations are carefully done and the results of optimized values are clearly presented. However, there are several concerns about the concept and results in the manuscript. The authors should address all the questions listed below in order to get further consideration about its publication in sensor. 1. The authors say that the proposed sensor can be easily integrated into the optical circuits. More information is necessary to explain how this be done. A system description, working principle, source and detector coupling with the sensing head should be included. 2. The work mainly deals with the trade-off between the evanescent field ratio and the propagation length. Simulation results show that they are approximately 37% and 880 μm, 22% and 292 μm for the two structures. Is it possible to estimate or comment on the limit of detection (LOD) for CO2 gas detection with the structure in a typical sensing system? 3. line 129, Figure of Merit (FOM) is a tradeoff parameter. Is the definition reasonable for high performance of the sensor based on beer-lambert law? 4. line 138, How could a 140nm layer mechanical strong enough to support the structure? Or is the structure only used for optimizing the parameters of the symmetric structure? 5. line 148 rail width >1000nm, slot height > 500nm, no remarkable influence on LSPP. The conclusion needs comments to explain. Simulations are focused at the wavelength of 4.26um for CO2 gas detection. Comments on application at a wavelength other than 4.26um are helpful. 6. line 198, For a Si3N4 thickness of 100 nm, an EFR of 42% and a LSPP of 1 mm are feasible. the thickness is too thin to support the structure? Is there any suggested fabrication technique? In the first half of the manuscript, all the optimal values are obtained in the symmetric structure. They are kept in the following simulation in asymmetric structure. Are the optimal values still good in the asymmetric structure? 7. ling 196, Figure 7 shows that the thinner the Si3N4 layer, the higher the FOM. It means that the structure with no Si3N4 layer (just two gold rod) is the best? Figure 8 also shows that n_sub=1 is the best choice, meaning no sub is the best? 8. The 100nm Si3N4 layer is a subwavelength delivery for the 4.26um wavelength delivery. Comments are also needed.Author Response
Please see the attachment.

Reviewer 2 Report
In this work, the authors described about the utility of plasmonic slot waveguides as a platform for CO2 sensing in the mid-IR. The paper is well written and presented. However, in my opinion, the work here does not represent sufficient novelty or uniqueness, neither in terms of design perspective, novel physics, nor new applications. The justification for using plasmonic slots considering the fact that they have such a high loss as compared to dielectric waveguides is not clear.
There are much better and beautiful demonstrations of waveguide based CO2 sensing including papers from the authors' group themselves (e.g. 10.1109/JPHOT.2018.2866628, https://doi.org/10.1016/j.sna.2018.05.013, https://doi.org/10.1364/OL.45.000109) If one calculates a similar FOM for these waveguide designs, one would get significantly higher values than those presented in the current paper.
Unless these major issues related to the importance and motivation of the work is addressed, I cannot recommend the paper for publication.
Below are some additional comments about other aspects of the paper.
1) References to some recent publications on mid-IR waveguide based CO2 sensing are missing such as ( https://doi.org/10.1364/OL.45.000109, and also some hollow waveguide gas sensors, https://doi.org/10.1021/acs.analchem.0c01586, https://doi.org/10.1007/s00216-011-5524-z )
2) I think the term symmetric structure for the waveguides with no silica substrate is not entirely justified. With the presence of 140 nm thick Silicon nitride with gold thicknesses ranging from 100nm to more, the silicon nitride is not thin enough to be negligible and the structure is not entirely symmetric. Other terminology such as free standing would be more appropriate.
3) It must be noted that while EFR is suitable for denoting the evanescent field fraction, for sensing purposes, confinement factor (Γ) which includes the contribution of the group velocity in addition to the evanescent field ratio correctly describes the sensing performance in the context of the Lambert Beer's law, which can even exceed 100%. ( Please see: https://doi.org/10.1038/s41377-021-00470-4, https://doi.org/10.1023/A:1007082915894 ). Thus, using Γ would be a more appropriate quantity for calculating the figures of merit instead of η.
4) What is the maximum field enhancement that is achieved in the vicinity of the plasmonic structures? With a such strong field enhancement that one typically gets from the plasmonic structures, would it cause any saturation of the CO2 absorption lines near 4.26 um ?
Reviewer 3 Report
In this manuscript, the authors presented the design flow of two hybrid plasmonic slot waveguides, and compared their figure of merits with respect to the geometric parameters as well as the substrate materials. The product of the evanescent field ratio and the propagation length is of interest to the performance of the design. While the authors showed the design considerations in detail, this theoretical work is merely based on simulations with simplified model. It would be of much more interest if the authors can generalize their design methodology into formulas, so that other researchers can refer them in their future work rather than relying on commercial software to do simulations. A step change would also put the manuscript in better shape, such as fabrication or real device testing. In addition, plasmonic waveguides have been extensively studied for gas sensing. In this work, there lacks a comparison of the current achievement with previous works. The authors should emphasize the novelty of this work and how it can contribute the field. The authors may also want to take the following comments into consideration.
- The propagation length of the SPP mode is based on the mode attenuation which is caused by the material loss of gold. This can be oversimplified. In fabrication, gold cannot be easily attached to dielectrics, and a titanium layer is usually used as an adhesive layer. In addition, the propagation loss can be much higher due to the scattering loss caused by roughness and high mode confinement.
- The manuscript has several plots of Real(Neff). It is unclear how the Real(Neff) benefits or help the design methodology?
- Only the upper cladding is counted as the gas sensing region, which could be a waste of the symmetric design. In fact, suspended membrane gas sensors should have better light-matter interaction since it can also sense the lower cladding region.
- Can the authors explain why the EFR drops first and then increases with the rail width in Figure 3 (a)?
- The language could be polished. There are some grammar and spelling errors, such as “On the one hand” in the abstract, “have been widely using” in the introduction.
Round 2
Reviewer 2 Report
I would like to thank the authors for addressing the points with due diligence and improving the manuscript. They have made the case for using plasmonic waveguides particularly due to their high field intensity, small mode volume and ability to have the field present in the low index region. While these are definitely interesting benefits for a plasmonic waveguide, I think another major benefit of using plasmonic waveguides could be through the use of surface functionalization or coating which ensures that the high field intensity interacts closely with the analyte of interest to give very high sensitivity (Please have a look at these papers: https://doi.org/10.1109/JQE.2019.2946839 ,https://doi.org/10.1021/acs.nanolett.8b03156 ). Some comments or discussions regarding this would be beneficial to elaborate on the potential applications and advantages of plasmonic waveguide based sensors.
Reviewer 3 Report
The authors have addressed my comments and questions, and the revision manuscript is in a better shape to be accepted.
Author Response
We would like to thank the reviewer for confirming our manuscript to be published in Sensors Journal.